# Psychotic Disorders in the Course of SARS-CoV-2 Infection or Uncomplicated Amantadine Treatment?—Case Report

**DOI:** 10.3390/ijerph192315768

**Published:** 2022-11-27

**Authors:** Dominika Tatar, Krzysztof Świerzy, Michał Błachut, Karina Badura Brzoza

**Affiliations:** Clinical Department of Psychiatry, Faculty of Medical Sciences in Zabrze, Medical University of Silesia in Katowice, 40-055 Katowice, Poland

**Keywords:** SARS-CoV-2, psychosis, amantadine, COVID-19

## Abstract

The mental health impact of SARS-CoV-2 infection is currently the subject of intense research. Mental disorders in the course of coronavirus infection are non-specific. They most often have a sudden onset and short-term course and resolve spontaneously or after the administration of low doses of antipsychotic drugs. At the same time, attempts have been made to develop recommendations for COVID-19 therapy. Single reports suggest the effectiveness of amantadine in the treatment. The mechanism of action of the drug in this case is not known; it is expected that amantadine, by reducing the expression of the cathepsin L gene, may interfere with SARS-CoV-2 replication. In addition, this drug stimulates dopaminergic transmission, which may result in numerous side effects, often of a neuropsychological nature, the most common of which are visual hallucinations. Therefore, it is extremely difficult to unequivocally diagnose the cause of mental disorders among patients with SARS-CoV-2 infection who took amatatide for off-label treatment. A clear assessment of whether the psychological symptoms in this group of patients are the primary or secondary clinical manifestation of the infection or a complication of amantadine treatment is difficult. In this context, we attempted to describe a case of a patient with psychotic symptoms who was confirmed with SARS-CoV-2 infection and treated with amantadine.

## 1. Introduction

The mental health impact of SARS-CoV-2 infection is currently the subject of intense research. The pathomechanism of central nervous system involvement in COVID-19 is still not fully understood; it is assumed that infection may occur through the ethmoid plate of the ethmoid bone near the olfactory bulb [1,2], then via the blood derivative or retrograde neuronal route to the CNS [3,4], and finally, through the expression of the angiotensin-converting enzyme receptor (ACE2) (identified as a functional SARS-CoV-2 receptor), present in many human organs, including in the nervous system and skeletal muscles, resulting in neuropsychiatric symptoms [4]. Due to the multiplicity of clinical symptoms, as well as clinical manifestations or consequences, it is difficult to diagnose which ailments are primary disorders and which are a symptom secondary to metabolic disorders or treatment (Table 1). Due to the diverse clinical picture, the diagnosis of COVID-19 is based primarily on the molecular study of the polymerase chain reaction in real-time (rRT-PCR); however, infection may also be suggested by deviations in routine diagnostic tests (Table 2).

Mental disorders in the course of coronavirus infection are nonspecific and include disturbances of consciousness, cognitive impairment, psychotic symptoms in the form of delusions, visual and auditory hallucinations, insomnia, affective lability, depressed or elevated mood, anxiety, anxiety and behavioral disorders, or catatonia [1,7,15,16]. Most often, they resolve after a sudden onset (developing within 48 h) [16], have a short course (about 2 weeks) [1], or resolve spontaneously or due to the use of antipsychotics (e.g., aripiprazole, risperidone, olanzapine)—often in low doses [1,16]—and benzodiazepine drugs [16]. Treating SARS-CoV-2 infection remains a challenge. There are attempts around the world to develop treatment guidelines based on clinical trials. Single reports [17,18,19] suggest the effectiveness of amantadine in the treatment of COVID-19. Amantadine has been used for the treatment of Parkinson’s disease, and there is observational evidence that patients infected with SARS-CoV-2 and taking amantadine as a treatment for Parkinson’s disease did not develop symptoms [20]. The mechanism of action of the drug in this case is unknown. Based on the in vitro studies conducted so far, it is assumed that amantadine, by reducing the expression of the cathepsin L gene, may impair the replication of SARS-CoV-2 [21,22]. In addition, it should be remembered that this drug stimulates dopaminergic transmission by stimulating the release of dopamine from presynaptic terminals, inhibiting its reuptake, stimulating D receptors, blocking the ionotropic NMDA receptor, and cholinolytic effects, which result in numerous side effects, often including neuropsychiatric symptoms, such as: sleep and mood disorders, exogenous paranoid psychoses with visual hallucinations, difficulties in concentration, as well as somatic symptoms of anxiety in the form of, among others, palpitations or sweating. It is believed that the release of dopamine indirectly through antagonism of the N-methyl-d-aspartate (NMDA) receptor is responsible for the occurrence of psychotic side effects [23,24,25,26,27,28].

Therefore, it seems extremely difficult to unambiguously diagnose the cause of mental disorders in patients with SARS-CoV-2 infection. A clear assessment of whether the psychological symptoms in this group of patients are the primary or secondary clinical manifestation of the infection or a complication of amantadine treatment is difficult to assess. In this context, we made an attempt to describe a case of a patient with psychotic symptoms who was confirmed to be infected with SARS-CoV-2 and treated with amantadine.

## 2. Case Report

A 47-year-old patient was admitted to a psychiatric ward from the Hospital Emergency Department (HED) due to his deteriorating mental state for about two weeks. The clinical picture was dominated by anxiety, mood swings, sleep disturbances, and delusional statements. The respondent emphasized that he feels watched and followed, that he has the impression that someone is threatening him and his family. The interview showed that a few days before the change in mental status, the patient developed flu-like symptoms with significant weakness and concentration disorders. Probably due to these symptoms, the patient fainted, which resulted in a bludgeoning wound, which was treated in the emergency department. The CT scan of the head performed at that time showed that there was a hematoma in the soft tissues of the right parieto-occipital area and inflammatory changes in the rump and sphenoid sinus. The conducted neurological examination did not reveal any deviations from the normal condition. The patient did not require hospital treatment and returned home. In the following days, the flu-like symptoms worsened, dyspnea and cough appeared, and the Primary Health Care physician (PHC) diagnosed pneumonia and started clarithromycin treatment, which lasted 7 days. Due to the deteriorating somatic condition manifested by increasing breathing problems, the patient was consulted in an internist emergency room. In the physical examination of the pulmonary fields, there were paralysis R > L groans, sinus tachycardia 120/min, and SpO2 95% without oxygen therapy. In the laboratory tests of arterial blood gas signs of hyperventilation, the CRP concentration was 6 mg/L, otherwise without significant deviations. A CT scan of the head was performed again, which showed no post-traumatic changes, while the chest X-ray PA and the lateral one revealed streaky and speckled densities in the lung area, most clearly visible in the lower part of the right lung and the middle of the left lung. SARS-CoV-2 infection was hypothesized, and an RT-PCR-SARS-CoV-2 test was performed. The result was negative. In addition, chest CT angio-CT examination ruled out acute pulmonary thrombotic embolism, while it showed numerous interstitial parenchyma densities of both lungs with areas of the milk glass type, with a visible localization predilection to the peripheral parts and involvement of the left upper, middle, and lower parts on the left side and right, mainly middle and inferior, sparing subpleural lung fragments. No fluid in the pleural cavities was visualized; however, enlarged lymph nodes were found under the bifurcation of the trachea in the short axis up to 10 mm (Group 7), in the area of the aorto-pulmonary window up to 11 mm (Group 5), and in both cavities up to 12 mm (Group 10). There was a pulmonary consultation; based on the radio-clinical picture, a hypothesis was made about a continuing or recent COVID-19 infection.

During the next internal and neurological consultation, no significant deviations from the normal condition were found. In view of the significantly deteriorating mental state with increasing anxiety and psychotic symptoms, it was decided to hospitalize the patient in a psychiatric ward. In the psychiatric examination upon admission to the ward, clear awareness, full autopsychic and allopsychic orientation, limited logical contact, and answers adequate only to simple, perfunctory questions, requiring inquiries and guidance with frequent disturbance of the way of thinking were found. There were visible vegetative symptoms of anxiety in the form of tremors of the whole body, sweating, and periodic hyperventilation. The patient expressed delusional, persecutory, and impoverished content, denied suicidal thoughts and intentions, and did not appear to be hallucinating. He was completely uncritical about his health. The objective interview showed that he had not been somatically ill before. He never underwent psychiatric treatment, and there were no mental illnesses in his immediate family. Additionally, the patient’s use of alcohol and psychoactive substances was denied. Until the first symptoms of the change in mental state appeared, the family did not notice any health problems. He functioned properly, and he worked professionally. In the first days of his stay in the ward, the patient presented severe anxiety, concentration and attention disorders, as well as insomnia. In pharmacotherapy, p.o. haloperidol was used with good results, up to 5 mg/d, lorazepam p.o. up to 7.5 mg/d, as well as parenteral hydration—5% glucose, 0.9% NaCl. Psychotic symptoms decreased; nevertheless, the patient showed ambivalent criticism towards the delusional content previously expressed. The psychomotor slowness and muscle tremors were still visible. The consulting neurologist noted a slightly unsteady gait and postural tremors in the upper limbs, which were inconsistent. Therefore, brain MRI was performed with contrast, which showed no abnormalities, and EEG was performed, the recording of which was within the normal range. Furthermore, the results of laboratory tests showed no significant deviations from the normal state, except for alanine hypertransaminasemia in the range of 66.3–73.1 U/L. In pharmacotherapy, p.o. haloperidol was used with good results, up to 5 mg/d, lorazepam p.o. up to 7.5 mg/d, as well as parenteral hydration—5% glucose, 0.9% NaCl. During the hospitalization, the patient admitted that, during the pneumonia treatment, despite obtaining a negative RT-PCR-SARS-CoV-2 result, he had tested the level of anti-SARS-CoV-2 IgM antibodies. Due to the obtained result of 32.4 S/CO (≥1.00 positive, reactive result), indicating active COVID-19 infection, without consulting a doctor, he started using amantadine at a dose of 200 mg/day for 5 days. In view of the obtained information, the level of anti-S SARS-CoV-2 S1-RBD IgG neutralizing antibodies was determined, the result of which was 2181.9 BAU/mL (≥7.1 positive, reactive result), confirming the presence of infection [9]. As a result of the pharmacological treatment, the psychotic symptoms and anxiety completely resolved, and a significant improvement in the previously observed slowdown and concentration disorders was achieved. The conducted psychological diagnostics did not reveal any pathologies in the personality profile. Upon discharge, the patient remained fully oriented, calm in behavior, in a balanced mood and drive, did not utter delusional content, did not hallucinate, and was fully critical of the previously occurring symptoms. However, trace, residual cognitive symptoms persisted—mainly in terms of abstraction and delayed recall; the Montreal Cognitive Assessment (MOCA) test corresponded to 25/30 points [29,30]. The patient was discharged from the ward on the 24th day of hospitalization with the final diagnosis: organic psychotic disorder (F06.2).

## 3. Discussion

SARS-CoV-2 infection is an important biological risk factor for the development of mental disorders. The pathogenetic mechanisms can be diverse. Mental disorders can be a direct consequence of brain damage from CNS infection or the body’s indirect immune response. They can also be the result of the applied treatment, as well as increased psychosocial stress (Moreira et al., 2021) [31]. Research shows that the tendency to be neuro-invasive is a common feature of coronaviruses [29]. Research on the tropism of SARS-CoV-2 shows that its penetration into human cells takes place mainly through the aforementioned ACE2 receptors, the presence of which has also been confirmed in the structures of the CNS. In animal models, the expression of the above-mentioned receptor (Wang et al., 2016) was shown in the amygdala, which is part of the limbic system. On this basis, it can be assumed that disorders related to the dysregulation of the limbic system may be caused by the presence of the virus itself in this area of the brain [30]. In addition, the functioning of the limbic system may also be impaired as a result of the stimulation of the immune system by SARS-CoV-2. Increasing the secretion of pro-inflammatory factors and activation of the kynurenine pathway of tryptophan metabolism may, in turn, be associated with the pathogenesis of psychiatric disorders, including psychoses, bipolar disorder, depression, and suicide [32]. Currently, it is estimated that more than one-third of COVID-19 patients develop neuropsychiatric symptoms [33]. The most common are non-specific neurological symptoms, such as delirium, cerebrovascular complications, encephalopathy, neuromuscular disorders, loss of smell and taste, as well as symptoms of depression, anxiety disorders, post-traumatic stress disorders, and residual executive function disorders [34]. According to Dinakaran et al. (2020), 0.9–4% of people infected with SARS-CoV-2 suffer from psychotic spectrum disorders [35]. In the described case, psychotic symptoms appeared and increased in a short period of time, accompanying para-flu symptoms, suggesting infection with SARS-CoV-2. The imaging examination of the lungs also showed changes indicating an active or past infection with the virus mentioned above, even though the PCR test was negative. The clinical picture of psychotic disorders could suggest the beginning of the schizophrenic process. However, due to the short duration of the episode, the temporal relationship with SARS-CoV-2 infection, the absence of pivotal symptoms typical of the schizophrenic process, and the achievement of rapid improvement after the introduction of pharmacotherapy, an external factor seems to be the more likely cause of the described disorders. Taking into account the results of IgM and IgG anti-SARS-CoV-2 immunoglobulins, the results of tests carried out in the department, as well as the clinical picture, it seems that the exogenous factor could be a recent SARS-CoV-2 infection. In addition, the cognitive difficulties observed in the patient at the end of hospitalization may be related both to the recent psychotic state and may also be the result of COVID-19. In the course of the above-mentioned disease, cognitive impairment has been reported, which may persist for 6 months to even 2 years as a result of the systemic inflammatory process and the affinity of the virus for the CNS [36,37]. Deterioration of cognitive functioning may concern its various aspects and manifest itself in weakening of fresh memory, attention function, and the so-called executive functions (including difficulties with inhibiting reactions, planning and solving problems, weakening cognitive flexibility) [37,38]. The role of amantadine in the development of psychotic symptoms cannot be excluded, due to its agonism towards D receptors, and thus the risk of inducing psychotic symptoms. Nevertheless, the most common side effects of amantadine therapy include psychotic symptoms in the form of hallucinations, mainly visual, absent in the described patient [24,25,26,27].

## 4. Conclusions

Therefore, it remains an unequivocally unexplained question what was the source of the destabilization of the patient’s mental state. The literature shows that people who have experienced COVID-19 have about twice the chance of developing a first-time mental disorder and a much higher rate of psychiatric disorders [39]. Descriptions of cases of severe mental disorders associated with SARS-CoV-2 are being presented more and more often, but only further studies will provide reliable information on the scale of this phenomenon.

## Figures and Tables

**Table 1 ijerph-19-15768-t001:** Selected symptoms of SARS-CoV-2 infection [4,5,6,7].

System	Symptoms
respiratory	shortness of breath, runny nose, cough, changes in the sense of smell, acute respiratory disorder (ARDS), pneumonia
cardiovascular	abnormal heart rhythm (bradyarrhythmias, tachyarrhythmias), acute heart failure, heart attack, cardiomyopathies, disseminated intravascular coagulation (DIC), thrombosis
digestive	nausea and vomiting, loss of appetite, diarrhea, abdominal pain
urinary	acute kidney injury (AKI), diffuse acute tubular injury, tubular necrotic changes
reproductive	damage to the testes and ovaries, hormonal disorders, in pregnant women: perfusion disorders of the fetal blood vessels, intrauterine growth inhibition (IUGR), premature rupture fetal membrane (pPROM) and preterm labor
skin	hives, itching, changes in the form of spots, pimples, papules, confluent rashes
neurological	headache and dizziness, delirium syndromes, strokes (ischemic, hemorrhagic), convulsions, encephalopathies, consciousness disorders, cognitive disorders, meningitis, ZOMR, ataxia, parkinsonism, photophobia and phonophobia

**Table 2 ijerph-19-15768-t002:** Selected abnormalities in additional tests in the course of SARS-CoV-2 [4,5,6,7,8,9,10,11,12,13,14].

Type of Examination	Abnormalities
Laboratory tests	↑ WBC, PLT, ALT, AST, urea, LDH, creatinine, CRP, PCT, ESR, D-dimers, CK-NAC ↓ WBC, lymphocytes, eosinophils, albumin
Antigen test	positive score
RT-PCR SARS-CoV-2	positive score
Anti-SARS-CoV-2 IgM coronavirus	reactive score
Chest X-ray	interstitial pneumonia, focus of turbidity
Chest CT scan	consolidations, compaction within the lung parenchyma, the image of “foggy/milky glass” or “snow blizzard”
Angio-CT of pulmonary arteries	acute pulmonary embolism, changes in the lung parenchyma
Chest ultrasound	pleural effusion, pneumonia
echocardiography	acute heart failure in the course of acute respiratory failure

## Data Availability

Not applicable.

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
