# Peer review of "Psychotic Disorders in the Course of SARS-CoV-2 Infection or Uncomplicated Amantadine Treatment?—Case Report"

_ijerph, 2022, doi:10.3390/ijerph192315768_

Round 1
Reviewer 1 Report
In the article entitled "Psychotic Disorders in the Course of SARS-CoV-2 Infection or Uncomplicated Amantadine Treatment?-Case Report.
The authors describe a patient with signs of anxiety and delirium. CT scan showed a soft tissue hematoma in the parieto-occipital region and he was treated with clarithromycin due to pneumonia. Two days earlier the patient received a consultation due to poor respiration, where RT-PCR results for SARS-CoV-2 were negative.
Some questions about this particular patient.
1. How long was the patient treated with clarithromycin?
2. How long did he take amantadine?
Regarding amantadine, there is scientific evidence from studies in clinical trials showing that amantadine is effective and safe as a treatment for SARS-CoV-2, I leave the reference: Rejdak K, Fiedor P, Bonek R, Goch A, Gala-Błądzińska A, Chełstowski W, Łukasiak J, Kiciak S, Dąbrowski P, Dec M, Król ZJ, Papuć E, Zasybska A, Segiet A, Grieb P. The use of amantadine in the prevention of progression and treatment of COVID-19 symptoms in patients infected with the SARS-CoV-2 virus (COV-PREVENT): study rationale and design. Contemp Clin Trials. 2022 May;116:106755. doi: 10.1016/j.cct.2022.106755. Epub 2022 Apr 4. PMID: 35390511; PMCID: PMC8978450.
There are also several observational studies for the treatment of SARS-CoV-2 infection.
Amantadine has been used for treatment of Parkinson's disease and there is observational evidence that patients infected with SARS-CoV-2 and taking amantadine as treatment for Parkinson's disease did not develop symptoms.
There is evidence that COVID-19 infection causes stroke and pulmonary embolism, I leave the reference: Soliman S, Ghaly M. Ischemic Stroke and Bilateral Pulmonary Embolism in COVID-19: COVID-Associated Coagulopathy or Heparin-Induced Thrombocytopenia. J Hematol. 2022 Feb;11(1):40-44. doi: 10.14740/jh956. Epub 2022 Feb 26. PMID: 35356633; PMCID: PMC8929198.
Stroke and pulmonary embolism in moderate CoViD-19
Arioli, D.; Varini, S.; Fabbri, S.; Maffei, C.; Cirino, S.; Angelini, M.; Verardo, A.; Romagnoli, E.; Violi, E.; Brugioni, L..
Italian Journal of Medicine ; 15(3):12, 2021.
In this particular patient the coronavirus may have caused the soft tissue hematoma, which led to neuroinflammation that may have generated the delirium and hallucinations.
The patient's pneumonia was treated, but not the neuroinflammatory process.
There are studies that show that COVID-19 infection produces changes in the structure of the brain that can lead to cognitive changes, I leave the citation: Douaud G, Lee S, Alfaro-Almagro F, Arthofer C, Wang C, McCarthy P, Lange F, Andersson JLR, Griffanti L, Duff E, Jbabdi S, Taschler B, Keating P, Winkler AM, Collins R, Matthews PM, Allen N, Miller KL, Nichols TE, Smith SM. SARS-CoV-2 is associated with changes in brain structure in UK Biobank. Nature. 2022 Apr;604(7907):697-707. doi: 10.1038/s41586-022-04569-5. Epub 2022 Mar 7. PMID: 35255491; PMCID: PMC9046077.
The case of this patient is interesting, and more studies should be done on the effect of SARS-CoV-2 on the brain and the cognitive changes that these can generate.
Regarding amantadine, the studies that have been carried out are promising in the treatment and prevention of COVID-19 infection.
It is a safe drug that has been used since 1969 for psychiatric treatment, anxiety, depression, and it has been estimated that in the long term it generates neuroplasticity in injured brains.
Author Response
Dears Sirs
Thank you for revising article entitled “Psychotic disorders in the course of SARS-CoV-2 infection or uncomplicated amantadine treatment? - Case report.” We have reorganized the paper following Reviewers’ comments. All changes in the text of article have
been written in red pencil. There are our responds to the Reviewers.
1. Patient was treated with clarithromycin under the care of a Primary Health Care physician due to pneumonia with accompanying dyspnea for 7 days.
-
He did take amantadine at a dose of 200 mg / day for 5 days.
- We introduced the suggested changes to the text of the work together with the literature justifying them.

Reviewer 2 Report
It is an intersting issue but there are a lot of concerns, since the patient is 47 years old and other pathological conditions should be taken into consideration before assumming that Covid-19 is responsiple for the psychotic episode.
The authors could dercribe more thoroughly the psychotic symptoms on the admission rather than mentionig only "xobic-persecutory delutions".
The personality of the patient should also be considered.
There are a lot of spelling mistakes and there are a lot of big periods without a fullstop that confuse the reader and at the end of the sentence the meaning is lost.
For instance
1. The second fullstop in the introdustion is after 7 rows!
2.r 38 it would be better to write "they resolve"
r.39 please make a new paragraph
r.45- r.51 small sentences to be easier understood
r.95 : normal state rather than condition
r.96: examination instead of study
r107-r109: "and EEg was perfomed"
The whole discussion is one paragraph! It needs to be rewritten.
Author Response
Dears Sirs
Thank you for revising article entitled “Psychotic disorders in the course of SARS-CoV-2 infection or uncomplicated amantadine treatment? - Case report.” We have reorganized the paper following Reviewers’ comments. All changes in the text of article have been written in red pencil. There are our responds to the Reviewers.
1. We made a detailed analysis of the patient's health condition using all the information we had. 2. The description of the mental state was supplemented upon admission to the psychiatric ward. 3. Linguistic errors were corrected while improving the style of work to make the text easier to read. 4. Linguistic proofreading was made with the help of a native speaker. 5. Corrections were made in the thesis discussion.
